# UDA: A Benchmark Suite for Retrieval Augmented Generation in Real-world Document Analysis

**Yulong Hui**
Tsinghua University
`huiyl22@mails.tsinghua.edu.cn`

**Yao Lu**
National University of Singapore
`luyao@comp.nus.edu.sg`

**Huanchen Zhang**[*]
Tsinghua University
`huanchen@tsinghua.edu.cn`

## Abstract

The use of Retrieval-Augmented Generation (RAG) has improved Large Language Models (LLMs) in collaborating with external data, yet significant challenges exist in real-world scenarios. In areas such as academic literature and finance question answering, data are often found in raw text and tables in HTML or PDF formats, which can be lengthy and highly unstructured. In this paper, we introduce a benchmark suite, namely Unstructured Document Analysis (UDA), that involves 2,965 real-world documents and 29,590 expert-annotated Q&A pairs. We revisit popular LLM- and RAG-based solutions for document analysis and evaluate the design choices and answer qualities across multiple document domains and diverse query types. Our evaluation yields interesting findings and highlights the importance of data parsing and retrieval. We hope our benchmark can shed light and better serve real-world document analysis applications. The benchmark suite and code can be found at `https://github.com/qinchuanhui/UDA-Benchmark`.

## 1 Introduction

Large Language Models (LLMs) have achieved remarkable success yet still face limitations [16]. One of the key challenges is to grapple with external knowledge and previously unseen data, which is a common scenario in real-world applications such as enterprise search and data analysis. For example, a company may need to query its proprietary technique documents; a financial expert may need to extract insights from the latest corporate reports; and a research group may need to assimilate cutting-edge academic papers to guide their innovations. To overcome this challenge, retrieval-augmented generation (RAG) that incorporates relevant content from an external data source into the LLM generation procedure, has emerged as a promising approach [30].

As shown in Figure 1, a common RAG workflow involves the following procedures: 1) parse the external data and segment it into chunks; 2) embed the chunks into vectors and create indexes; 3) retrieve the most relevant chunks according to the user query, and 4) assemble the prompt with relevant chunks (i.e., context) as input for the LLM to generate the response. Recently, a multitude of advanced RAG techniques have been proposed with improved retrieval policies [3, 64], context chunk compression [59, 56], and pre-training strategies [49]. Furthermore, as an alternative approach to RAG, recent advances in long-context LLMs [8, 11, 62] have empowered querying directly on lengthy data without chunking and retrieval.

---

[*]Huanchen Zhang is also affiliated with the Shanghai Qi Zhi Institute. Corresponding author.

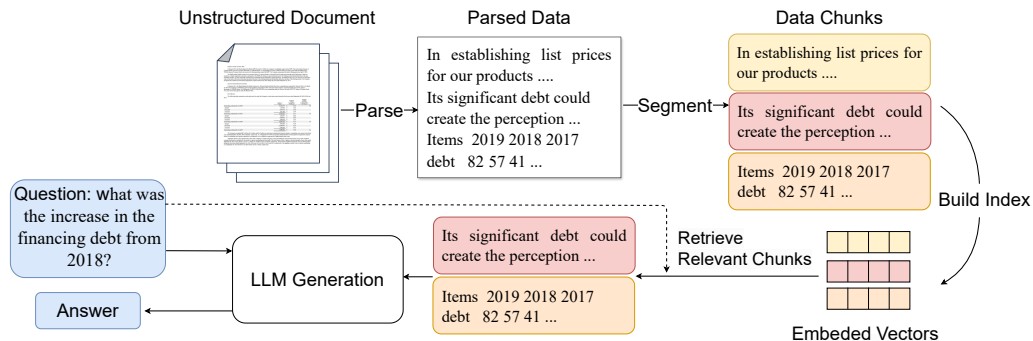

Figure 1: An example of basic RAG processing on unstructured documents.

According to a study by Forbes, 95% of business organizations in various domains such as finance and technology need analyzing unstructured texts and tables in raw documents like web pages and PDFs [6]. Analyzing unstructured documents in the world poses the following challenges:

*Unstructured inputs.* Parsing unstructured documents into regularized text and tables is error-prone. Unlike plain text, unstructured documents often contain intricate layouts and redundant symbols. Prior works [51, 63] have incorporated vision and language models, but their effectiveness is still doubtful. Further, multi-modal data, e.g., tables, require improved indexing and retrieval strategies because classic text embeddings disregard structural information from these data.

*Lengthy documents*, such as financial reports spanning hundreds of pages, necessitate effective embedding and retrieval mechanisms.

*Query answering strategies.* User queries span from extractive queries to complex arithmetic reasoning; each may require a different answering strategy, such as using Chain-of-Thought[57] or external tools like Code Interpreters [65]. We wonder how these design choices can impact the end-to-end query answering quality.

In this paper, we propose a benchmark suite that enables the evaluation of various components of RAG-based unstructured document analysis. Specifically, we leverage the Unstructured Document Analysis (UDA) dataset to cover finance, academia, and world knowledge with a total of 2,965 documents and 29,590 expert-annotated Q&A pairs. UDA enables end-to-end evaluations of diverse Q&As and granular analyses of individual components within the RAG pipeline.

Unlike prior datasets and benchmarks which often assume clean or segmented inputs [12, 35, 15], we exploit the design considerations in end-to-end document analysis. We performed extensive experimental analysis to cover different data extraction policies, retrieval and generation strategies, and a range of LLMs. We also compared RAG-based solutions with those that use LLMs with long context capabilities.

Our analysis leads to interesting findings. First, despite that various computer-vision- or language-based parsing techniques have been proposed, conventional solutions may fail to improve the overall Q&A quality due to irregular edge cases. We also found that smaller retrieval models could perform reasonably well in certain RAG applications, while Chain-of-Thought approaches improve the answer quality in zero-shot numerical document analysis. However, long-context LLMs often fall short in these tasks.

We will actively update the benchmark suite and incorporate more state-of-the-art RAG solutions and LLMs in our benchmark. We hope such efforts will shed light on future research and production of RAG- and LLM-based document analysis.

## 2 Related Work

**Retrieval Augmented Generation** Large Language Models (LLMs) demonstrate remarkable abilities but struggle with external knowledge and the latest unseen data. Retrieval Augmented Generation (RAG) addresses these limitations by incorporating external information to enrich LLMs' responses,

Table 1: Summary and comparison of Q&A datasets. *Raw documents*: datasets in the native file format, without extraction or parsing; *Long content*: the provision of unsegmented, un-retrieved long content.

| Dataset | Text + tables | Raw documents | Long content | Diverse Questions | Sources |
|---|---|---|---|---|---|
| HybridQA [10] | ✓ | | | ✓ | Wikipedia |
| WiKiTableQuestion [39] | | | | ✓ | Wikipedia |
| TriviaQA [24] | ✓ | ✓ | ✓ | ✓ | Wikipedia |
| VisualMRC [52] | ✓ | | | ✓ | Wikipedia |
| FinQA [12] | ✓ | | | | Finance |
| TAT-DQA [66] | ✓ | ✓ | | ✓ | Finance |
| Qasper [13] | ✓ | | ✓ | ✓ | Papers |
| PDF-VQA [14] | ✓ | ✓ | ✓ | | Medicine |
| DocVQA [35] | ✓ | ✓ | | ✓ | **Multiple** |
| NarrativeQA [26] | | | ✓ | ✓ | **Multiple** |
| UDA (Ours) | ✓ | ✓ | ✓ | ✓ | **Multiple** |

yielding more precise and credible outputs [44]. Furthermore, several innovative techniques have been developed to refine the procedure beyond the basic RAG approach [16]. For instance, Self-RAG [3] and FLARE [23] determine the retrieved content actively according to the generation results. LLMLingua [22, 21], RECOMP [59], and FILCO [56] focus on filtering and condensing the context input to enhance the information efficiency. Additionally, specialized pre-training and fine-tuning techniques can also optimize the process [31, 48, 49]. Despite these advancements, current approaches often overlook the complexities in real-world unstructured document analysis, such as unstructured data and table schemas, extensive document lengths, and diverse analytical queries.

**Prior Benchmarks for RAG** offer tools for assessing various dimensions of RAG. RGB [9] benchmarks RAG on robustness and negative rejection. CRUD-RAG [34] introduces a Chinese news dataset for multifaceted evaluation, including text continuation, question answering, and error correction. ALCE [15] evaluates the performance of generating cited responses. In contrast, our benchmark emphasizes the tasks of document comprehension and analysis.

**Prior Benchmarks for Q&A** often inadequately represent real-world scenarios. For example, mainstream datasets like TriviaQA [24], HotPotQA [61], SQuAD [42], and NaturalQuestions [29] predominantly utilize the Wikipedia sources that have limited application scope and potentially overlap with LLM's internal knowledge. Moreover, datasets such as QuALITY [38] and NarrativeQA [26] are pure plain text, while WikiTableQuestions [39] and SQA [18] are pure tabular data. They fail to capture the complexity of real-world analytical documents. Additionally, datasets like FinQA [12] and VisualMRC [52] present well-structured or segmented content directly, thus sidestepping the intricacies of parsing and retrieval. Table 1 summarizes the features of existing datasets and highlights the uniqueness of our UDA benchmark in real-world document analysis.

## 3 Dataset: UDA

In this section, we outline the composition and construction of UDA. Each data item within UDA is logically structured as a triplet $(D, q, a)$, where $D$ represents a complete unstructured document, $q$ denotes a question raised from the document, and $a$ signifies the ground truth answer (refer to the data example in Appendix A). To mirror the authenticity of real-world applications, the documents are retained in their original file formats without parsing or segmentation.

**Dataset Composition.** Our UDA dataset includes six sub-datasets across three pivotal domains: finance, academia, and knowledge bases, reflecting typical use cases in document analysis. As delineated in Table 2, the dataset spans table-based and text-based (or hybrid) QA formats in each domain to ensure that the evaluation covers different data patterns. Moreover, UDA contains 2,965 documents with a wide range of content length and 29,590 expert-annotated Q&A pairs that vary from extractive queries to arithmetic reasoning (see examples in Table 3). These features profoundly embody the breadth and depth of practical real-world applications.

Table 2: An overview of sub-datasets in UDA and their statistics

| Domain | Sub Dataset | Doc Format | Doc Num | Q&A Num | Avg #Words | Avg #Pages | Tot Size | Q&A Types |
|---|---|---|---|---|---|---|---|---|
| Finance | FinHybrid | PDF | 788 | 8190 | 76.6k | 147.8 | 2.61 GB | Arithmetic |
| | TatHybrid | PDF | 170 | 14703 | 77.5k | 148.5 | 0.58 GB | Extractive, counting, arithmetic |
| Academic Paper | PaperTab | PDF | 307 | 393 | 6.1k | 11.0 | 0.22 GB | Extractive, yes/no, free-form |
| | PaperText | PDF | 1087 | 2804 | 5.9k | 10.6 | 0.87 GB | Extractive, yes/no, free-form |
| World Knowledge | FetaTab | PDF & HTML | 878 | 1023 | 6.0k | 14.9 | 0.92 GB | Free-form |
| | NqText | PDF & HTML | 645 | 2477 | 6.1k | 14.9 | 0.68 GB | Extractive |

Table 3: Examples of different Q&A types

| Q&A Types | Example Question | Example Answer |
|---|---|---|
| Extractive | Who has the longest win streak in MMA? | Anderson Silva |
| Yes/No | Are experiments performed with any other pair of languages? | No |
| Free-form | How did Hayden Panettiere fare at the 2012 and 2013 Golden Globes? | Hayden Panettiere received two nominations for the Golden Globe Award, Best Supporting Actress Series, Miniseries or Television Film, for her work on Nashville in 2012 and 2013. |
| Counting | How many regions have revenues of more than $20,000 thousand? | 2 |
| Arithmetic | What was the percentage increase in cash dividends from 2015 to 2016? | $(0.29 - 0.25) \div 0.25 * 100\% = 16\%$ |

**Label Collection.** We first collect the Q&A labels from the open-released datasets (i.e., source datasets), which are all annotated by human participants. Specifically, our **FinHybrid** is based on the financial numerical reasoning dataset FINQA [12], which is constructed based on the public earnings reports of S&P 500 companies. **TatHybrid** is derived from TAT-DQA [66], whose Q&A pairs are accompanied by the document snapshot of 1 to 3 pages from public financial annual reports. Both **PaperTab** and **PaperText** are based on Qasper [13], a reading comprehension dataset based on NLP research papers. **FetaTab** is built upon FetaQA [36], a question-answering dataset for tables from Wikipedia pages. **NqText** is derived from the widely used Q&A dataset, Natural-Questions [28], which uses the Wikipedia pages as context. Its questions are collected from the Google Search engine, and the answers are human-annotated.

**Dataset Construction.** We conduct a series of essential constructing actions after collecting the Q&A labels from the source datasets. The integrity of original documents is crucial for the fidelity of document analysis. However, most of the source datasets only offer well-parsed and segmented partial content without the complete document. To address this problem, we perform a comprehensive source-document identification process, including retrieval, verification, and cleaning. (1) Retrieval: the original documents are sourced from certain platforms, such as Wikipedia [58] and arXiv [2]. We conduct retrieval on these platforms using the metadata from the source datasets, such as titles, timestamps, and document types. (2) Verification: we verify the accuracy of the retrieved document by cross-referencing the content fragments in existing datasets, collecting only exact matches. Specifically, we employ the pypdf [41] library for automatic data parsing and comparison, supplemented by manual inspection. (3) Cleaning: we remove the documents that are inaccessible or not available, such as damaged files and withdrawn papers.

Then we embark on a rigorous matching and reorganization effort, enhancing the data quality and forming complete triplet data pairs, i.e., document-question-answer. This involves the following transformations: (1) data cleaning by removing the Q&A pairs or documents lacking essential answers; (2) standardizing diverse data formats into consistently structured tables and JSON files, addressing the heterogeneity of presentation across different sub-datasets; (3) categorizing the queries in Qasper into table-centric and text-centric, thus forming the PaperTab and PaperText subsets for different application patterns; (4) converting HTML-token-based data type into natural language, forming our user-friendly NqText dataset (5) manually annotating some tables within Qasper to serve as references for evaluating parsing strategies.

Table 4: The structure of the sampled datasets used in the following evaluation.

|  | Sum | FinHybrid | TatHybrid | PaperTab | PaperText | FetaTab | NqText |
|---|---|---|---|---|---|---|---|
| # Docs | 1201 | 100 | 150 | 307 | 194 | 300 | 150 |
| # Q&A pairs | 2503 | 451 | 450 | 393 | 480 | 350 | 379 |

## 4 Benchmarks and Evaluations

We provide a systematic benchmark of various modules in a typical RAG workflow, as well as an end-to-end LLM-based evaluation. The focused items of our benchmark include:

- The effectiveness of various table-parsing approaches, including raw-text extraction, computer vision (CV)-based, CV-LLM-based and advanced multi-modal parsing (Section 4.1).

- The performance of different indexing and retrieval strategies, spanning sparse retrieval, classic dense embedding, and advanced retrieval model (Section 4.2).

- The influence of precise retrieval on the quality of LLM interpretation (Section 4.2).

- The effectiveness of long-context LLMs compared to typical RAGs (Section 4.3).

- Comparison of different Q&A strategies, such as Chain-of-Thought reasoning and the integration of external code execution (Section 4.4).

- End-to-end comparisons of various LLMs across diverse applications (Section 4.5).

**Metrics**    To evaluate the quality of LLM-generated answers, we apply widely accepted span-level F1-score [42] in PaperTab, PaperText, FetaTab, and NqText datasets, where ground-truth answers are in natural language and the source datasets also utilize this metric. We treat the prediction and ground truth as bags of words and calculate the F1-score to measure their overlap. In financial analysis, the assessment becomes more intricate due to numerical values. For the TatHybrid dataset, we adopt the numeracy-focused F1-score, introduced by Zhu, et al. [67], which considers the scale and the plus-minus of numerical values. In the FinHybrid dataset, where answers are always numerical or binary, we rely on the Exact-Match metric but allow for a numerical tolerance of $1\%$, accounting for rounding discrepancies. Furthermore, we also incorporate the LLM-based method for a more comprehensive evaluation (more details in Appendix B.5). To assess the effectiveness of retrieval strategies, we identify the factual evidence in retrieved chunks using the relative length of the Longest Common Subsequence (LCS) [40] instead of the exact match, because extracted PDF data chunks often include extraneous symbols that can hinder exact matches while still containing crucial evidence.

**Experiment setups**    In our experiments, we evaluate the performance of various decomposed RAG components, mainly utilizing two representative LLMs: 1) GPT-4 [1], exemplifying the large-scale powerful model, proposed by OpenAI; 2) Llama-3-8B [53], representing the compact yet capable model, proposed by Meta. Furthermore, to ensure a thorough comparative analysis, we also include the end-to-end experiment encompassing a suite of additional LLMs: 3) LLama-3-70B [33], an open-source large-scale model; 4) Qwen-1.5-32B and Qwen-1.5-7B [5], introduced by Alibaba, notable for its 32k token context window; 5) Mixtral-8x7B [20], a Mixture-of-Experts model innovated by MistralAI; 6) Mistral-7B [19], also from MistralAI; 7) CodeLlama-7B and CodeLlama-13B [45], llama models tailored for code generation.

Following prior works in [4, 60, 68], we focus on zero-shot LLM generation, yet add an extra formatting example to align the output with the desired pattern (refer to Appendix B.1). For the GPT-4 model, we leverage the Azure-OpenAI API to access GPT4-Turbo-1106-Preview with the context window of 128k. Other open-source models are obtained from Huggingface, and we always use the instruct-tuned version. The inference is done on 4 NVIDIA-A100 GPUs. To reduce the compute costs, we randomly sample 1201 documents (in PDF format) accompanied by 2503 question-answer pairs to form our evaluation set (detailed in Table 4). We believe it serves as a practical performance indicator for real-world scenarios.

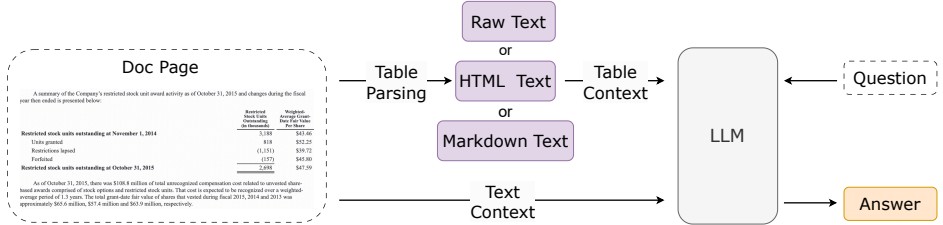

Figure 2: The procedure of the table parsing experiment

Table 5: Performance scores (EM or F1) of LLMs using varying parsing strategies on table-based Q&A tasks.

| Dataset | LLM Name | Well Parsed | GPT-4-Omni | Raw Text | CV | CV + LLM |
|---|---|---|---|---|---|---|
| Tabular FinHybrid (EM) | GPT-4-Turbo | 71.9 | **72.4** | 68.0 | 61.3 | 52.4 |
| | Llama-3-8B | **59.5** | 56.3 | 51.6 | 44.6 | 40.2 |
| PaperTab (F1) | GPT-4-Turbo | 42.8 | **44.3** | 42.4 | 38.6 | 40.7 |
| | Llama-3-8B | 35.8 | **37.7** | 36.5 | 34.6 | 32.1 |

## 4.1 Evaluating Data Parsing

We evaluate various parsing methods to extract tabular information from PDF files and analyze their influence on the downstream tasks, utilizing the table-based questions from PaperTab and FinHybrid (more details in Appendix B.2). As shown in Figure 2, each question is paired with a PDF page that contains the clue tables; doing so prevents inaccurate retrievals. The tabular data are parsed into text and merged with the rest of the text content as the input context to the LLM.

We evaluate several existing approaches of table parsing: (1) **Raw text extraction**, which employs a PDF text extractor [41] to extract all the characters. (2) Classic Computer Vision (**CV**) based approach, which often performs layout detection and OCR extraction at the same time. We follow [55] to use Yolox [17], Tesseract [50] and TableTransformer [51] models together. (3) **CV + LLM** method, which further employs an LLM to transform the outputs of (2) into Markdown tables. (4) For the advanced multi-modal approach, we employ the latest **GPT-4-Omni** [37] to convert image-based document tables into Markdown format. (5) The manually-verified **well-parsed** tables serve as the parsing ground truth.

Table 5 reveals that GPT-4-Omni outperforms other approaches, while surprisingly, raw text extraction also yields decent results. We found that the queried tables are relatively simple; the structural markers from the raw text, such as line-breakers and space, are often adequate for LLMs to understand the table. Classic CV methods, if not meticulously tuned, may struggle in handling non-standard table presentations, i.e., edge cases (see an example in Figure 3). Additionally, employing GPT-4-Omni directly for question-answering scores 69.8 and 35.4, lower than sequentially parsing and generating with GPT-4 (i.e., 72.4 and 44.3).

We also observe from the FinHybrid dataset that the GPT-4 model shows a modest 5.7% improvement with well-parsed data over raw-text data, while the much smaller Llama-3-8B offers a significant 15% enhancement, suggesting that compact models with a limited capability of parsing table layouts, may benefit more from enhanced parsing. In the PaperTab dataset, where completely accurate information is less critical, GPT-4-Omni and raw-text parsing could even outperform well-parsed tables by preserving structural cues that highlight important elements for LLM interpretation.

**Remark.** Evaluations here suggest (1) CV-based parsing methods may require adaptation for edge cases before they can be useful; (2) smaller LLMs may be impacted more by uncleaned input data, when requiring accurate and specific information.

|  | North America | Europe, Middle East & Africa | Asia Pacific | South America | Total |
|---|---|---|---|---|---|
| **Source PDF Page** | | | | | |
| Electrical/Electronic Architecture...... | 32 | 34 | 25 | 5 | 96 |
| Powertrain Systems............................ | 4 | 8 | 5 | 1 | 18 |
| Electronics and Safety........................ | 3 | 6 | 3 | — | 12 |
| Total............................ | 39 | 48 | 33 | 6 | 126 |

**Raw-Text Extraction**
North AmericaEurope,
Middle East
& Africa Asia Pacific South America Total
Electrical/Electronic Architecture...................... 32 34 25 5 96
Powertrain Systems............................ 4 8 5 1 18
Electronics and Safety........................ 3 6 3 — 12
Total............................ 39 48 33 6 126

**Classic-CV Parsed Table (in visible format)**

|  | North America | Europe, Middle East & Africa | Asia Pacific | South America | Total |
|---|---|---|---|---|---|
| Electrical/Electronic Architecture 32 | | 34 | 25 | 5 | 96 |
| Powertrain Systems. | | 8 | | 1 | 18 |
| Electronics and Safety. | | | | | |
| Total. | | | | | |

**GPT-4-Omni Parsed Table**

```
|                                    | North America | Europe, Middle East & Africa | Asia Pacific | South America | Total |
|------------------------------------|---------------|------------------------------|--------------|---------------|-------|
| Electrical/Electronic Architecture | 32            | 34                           | 25           | 5             | 96    |
| Powertrain Systems                 | 4             | 8                            | 5            | 1             | 18    |
| Electronics and Safety             | 3             | 6                            | 3            | —             | 12    |
| Total                              | 39            | 48                           | 33           | 6             | 126   |
```

Figure 3: An example of table parsing with different strategies. Raw-text-extraction preserves the informational content with structural markers; CV-based method may struggle with the irregular table presentation; GPT-4-Omni yields the highest accuracy.

Table 6: Relative LCS scores in different retrieval strategies. We evaluate the presence of evidence in most related 1, 5, 10 and 20 chunks.

| | Model | FinHybrid | | | | PaperTab | | | | PaperText | | | | FetaTab | | | | NqText | | | |
|---|---|---|---|---|---|---|---|---|---|---|---|---|---|---|---|---|---|---|---|---|---|
| | | @1 | @5 | @10 | @20 | @1 | @5 | @10 | @20 | @1 | @5 | @10 | @20 | @1 | @5 | @10 | @20 | @1 | @5 | @10 | @20 |
| Sparse | BM-25 | **65.6** | **83.7** | **87.4** | **90.0** | 46.0 | 79.7 | 90.0 | 92.3 | 47.4 | 80.0 | 88.0 | 89.9 | 68.3 | 91.9 | **95.2** | **96.2** | 42.0 | 69.2 | 75.8 | 80.3 |
| Dense Embedding | all-MiniLM-L6 | 49.1 | 71.8 | 78.2 | 84.0 | 51.3 | 81.7 | 90.4 | 92.9 | 45.7 | 76.7 | 85.9 | 89.9 | 63.6 | 90.8 | 94.4 | 95.3 | 49.1 | 71.1 | 77.3 | 80.7 |
| | all-mpnet-base | 48.7 | 74.7 | 81.7 | 86.7 | 50.2 | 82.2 | 90.8 | 92.9 | 40.8 | 75.3 | 86.3 | 89.8 | 66.3 | 91.5 | 94.6 | 95.5 | 50.3 | 73.4 | 78.8 | 81.7 |
| | OpenAI | 57.2 | 80.1 | 85.2 | 89.3 | **55.5** | **85.4** | **91.8** | **93.0** | **52.2** | **83.1** | **89.0** | **90.3** | **69.7** | **92.7** | 95.0 | 95.7 | **50.9** | **74.9** | **80.2** | **82.3** |
| Advanced | Col-BERT | 54.4 | 75.0 | 80.4 | 85.0 | 47.8 | 79.7 | 89.2 | 92.7 | 48.4 | 77.1 | 86.8 | 89.8 | 67.8 | 91.5 | 94.6 | 95.4 | 47.3 | 70.3 | 76.5 | 80.2 |

## 4.2 Indexing and Retrieval

We evaluate the performance of 5 different models under the following retrieval paradigms: 1) **BM-25** [7, 54], a lightweight sparse retrieval method without complex neural networks, ranking document segments based on the appearing frequency of query terms. 2) **all-MiniLM-L6** from SentenceTransformer [43], a prevalent dense embedding model, mapping sentences to a 384-dimensional dense vector space. 3) **all-mpnet-base**, another widely utilized embedding model from SentenceTransformer, noted for its larger architecture and improved performance. 4) **text-embedding-3-large model**, the latest embedding model from **OpenAI**, with enhanced capability. These classic dense embedding models process both query and document segments into vectors, and employ cosine similarity measures to retrieve the most relevant segments. 5) **ColBERT** [46], an advanced retrieval model, relying on token-level embedding and fine-grained contextual late interaction.

We use the relative length of the Longest Common Subsequence (LCS) to demonstrate the presence of human-annotated evidence in retrieved chunks (more experimental details in Appendix B.3). As shown in Table 6, OpenAI's text-embedding-3-large model excels in most datasets except for FinHybrid, where the simpler BM-25 approach intriguingly outperforms. This could be attributed to the fact that financial queries often contain more precise details, such as dates or keywords; this aligns well with the direct keyword-matching of BM-25.

We also conduct end-to-end experiments to verify the impact of the retrieval quality. We evaluate the answer quality with top-5 chunks retrieved using OpenAI embeddings or with the human-annotated evidential chunks. As shown in Table 7, providing more relevant context to LLMs improves the answers in most cases, particularly in arithmetic-reasoning tasks such as FinHybrid and TatHybrid. Interestingly, for the FinHybrid dataset, the Llama-3-8B model achieved a score of 51.0 when given accurate context, outperforming GPT-4 with model-retrieved chunks. However, for knowledge-based questions from the NqText and FetaTab, the answer quality remains less affected. This is because complex numerical reasoning demands more precise evidence for accurate arithmetic operations,

Table 7: End-to-end answer scores using retrieved and human-annotated context

| LLM Name | Context Type | FinHybrid | TatHybrid | PaperTab | PaperText | FetaTab | NqText |
|---|---|---|---|---|---|---|---|
| Llama-3-8B | OpenAI Retrieval @5 | 37.9 | 22.5 | **35.5** | 42.3 | 56.6 | **31.7** |
| | Human-annotated | **51.0** | **35.9** | 35.1 | **46.4** | **57.5** | 31.5 |
| | Improvement | 35% | 60% | -1% | 10% | 2% | -1% |
| GPT-4-Turbo | OpenAI Retrieval @5 | 45.9 | 43.5 | 40.3 | 45.8 | **61.5** | 37.4 |
| | Human-annotated | **69.4** | **57.7** | **42.0** | **56.5** | 59.5 | **39.0** |
| | Improvement | 51% | 33% | 4% | 23% | -3% | 4% |

Table 8: Performance scores between long-context and RAG mechanism.

| LLM Name | Input Type | FinHybrid | TatHybrid | PaperTab | PaperText | FetaTab | NqText |
|---|---|---|---|---|---|---|---|
| Qwen-1.5-7B | OpenAI Retrieval @5 | **21.0** | **26.6** | **31.4** | **39.1** | 58.1 | **32.4** |
| | Long Context | 3.0 | 20.9 | 26.3 | 33.1 | **58.7** | 30.2 |
| GPT-4-Turbo | OpenAI Retrieval @5 | **43.4** | **46.3** | **43.5** | 47.1 | 61.8 | **35.8** |
| | Long Context | 37.4 | 36.9 | 43.3 | **47.4** | **63.3** | 35.4 |

whereas LLMs can leverage a wide range of narrative information to derive answers to knowledge-based questions.

**Remark.** We found that the model scaling law may not hold true in retrieval scenarios. The retrieval quality does matter, particularly in arithmetic tasks, but the incremental benefit of including additional chunks diminishes. Some use cases (e.g., queries that involve exact entity matching) may prefer some specific data embedding or indexing mechanisms.

## 4.3 RAG vs. Long Context

We compare RAG-based methods with long-context LLMs, utilizing GPT-4-Turbo with a 128k context window and Qwen-1.5-7B with a 32k context window. Due to the high cost of long context inference, we conduct this experiment on a subset of 600 documents (more details in Appendix B.4). The results are demonstrated in Table 8.

**Remark.** We notice that for free-form or knowledge-based tasks (i.e., paper-based and wiki-based Q&A), RAG and long-context solutions demonstrate comparable capability. Conversely, in tasks with more numerical reasoning (i.e., financial Q&A), long-context LLMs fail to match the performance of RAG. In such use cases, the excess of verbose content might hinder long-context LLMs in pinpointing facts and performing numerical reasoning effectively (see an example in Table 10). Additionally, RAG is likely to surpass the long-context mechanism more in the smaller LLM, given its constrained capacity to handle large volumes of data.

## 4.4 Evaluating Chain-of-Thought and Code Interpreters

In real-world analytical queries, reasoning capabilities can be essential, yet LLMs face limitations in this area [25]. To overcome these constraints, advanced methods such as Chain-of-Thought (CoT) [57] and Code-Interpreter (CI) [65] have been introduced. CoT prompts LLMs to generate a series of intermediate reasoning steps, whereas CI lets the LLM produce executable codes and then invoke an external executor to derive the answer. In this section, we evaluate the efficacy of basic generation, the CoT approach, and CI methods using the numerical reasoning dataset FinHybrid. We use the top-5 chunks retrieved with OpenAI's embedding as the context and benchmark GPT-4-Turbo, Llama-3-8B, and the code-tailored CodeLlama models. The CoT approaches are implemented with step-wise instructive prompts, while the basic CI method asks LLMs to produce Python codes if necessary (more details in Appendix B.1).

**Remark.** As illustrated in Table 9, the Chain-of-Thought (CoT) approach outperforms others across all model configurations (see an example in Table 10). The Llama-3-8B model shows improvements when incorporating codes, yet CoT methods still prove superior. CodeLlama models, however, struggle with generating viable code in this scenario with lengthy context and ambiguous code-gen instructions. GPT-4-turbo exhibits a native ability to produce step-by-step explanations, leading to

Table 9: Exact-match scores of different generation strategies on the FinHybrid dataset

| LLM Name | Base | CoT | Code |
|----------|------|------|------|
| Llama-3-8B | 21.3 | **37.9** | 26.4 |
| CodeLlama-7B | 5.5 | **7.3** | 5.5 |
| CodeLlama-13B | 10.6 | **11.3** | 9.1 |
| GPT-4-Turbo | **45.9** | **45.9** | 32.2 |

Table 10: Case study of long-context and generating strategy. RAG outperforms the long-context through more accurate evidence retrieval, and CoT's superiority is attributed to the integration of explicit reasoning steps.

| Question | LLM Name | Strategy | LLM's Response |
|----------|----------|----------|----------------|
| What percentage of contractual obligations is due to maturities of long-term debt? | Qwen-1.5-7B | long-context | 33% ✗ |
| | | **RAG** | 690 million out of 1416 million, 49% ✓ |
| What percent of the share-based compensation expense was related to stock options? | Llama-3-8B | basic-gen | 100% ✗ |
| | | code | 94.4% (no code is generated, direct response) ✗ |
| | | **CoT** | To find the percentage, we can divide the portion related to stock options by the total share-based compensation expense ($7 million / $36 million) x 100 = 19.4% ✓ |

Table 11: End-to-end performance scores of different LLMs

| LLM Name | Avg | FinHybrid | TatHybrid | PaperTab | PaperText | FetaTab | NqText |
|----------|-----|-----------|-----------|----------|-----------|---------|--------|
| GPT-4-turbo | **45.7** | **45.9** | **43.5** | **40.3** | **45.8** | 61.5 | 37.4 |
| Llama-3-70B | 42.5 | 43.5 | 30.9 | 38.7 | 44.4 | 63.3 | 33.9 |
| GPT-3.5 | 40.9 | 36.6 | 33.9 | 35.5 | 42.1 | **64.1** | 33.1 |
| Qwen-1.5-32B | 38.9 | 31.3 | 27.9 | 31.6 | 43.1 | 58.4 | **41.3** |
| Llama-3-8B | 37.7 | 37.9 | 22.5 | 35.5 | 42.3 | 56.6 | 31.7 |
| Mixtral-8x7B-v0.1 | 34.5 | 28.4 | 22.5 | 35.0 | 38.1 | 54.1 | 29.1 |
| Qwen-1.5-7B | 33.8 | 17.0 | 22.6 | 28.0 | 37.7 | 58.6 | 38.9 |
| Mistral-7B-v0.2 | 26.6 | 18.2 | 15.9 | 20.9 | 22.7 | 54.3 | 27.3 |
| Llama-3-8B-NoRAG | 18.0 | 3.6 | 6.0 | 13.0 | 16.2 | 47.7 | 21.7 |
| GPT-4-turbo-NoRAG | 17.6 | 0.4 | 3.0 | 15.3 | 18.9 | 48.6 | 19.6 |

equivalent performance between basic generation and CoT methods. It is worth noting that while code interpreter has been demonstrated capable of handling numerical and tabular data [27, 68], its efficacy in document analysis is hindered by the unstructured tables and lengthy context, if just using the basic code strategy.

## 4.5 End-to-End Evaluations

Assembling the insights derived from the above analyses, we construct an end-to-end RAG pipeline. Specifically, we parse unstructured data from PDFs leveraging the raw-text extraction approach, and employ OpenAI's text-embedding-3-large model to index and retrieve relevant data chunks. In the generation phase, we incorporate the Chain-of-Thought approach to handle arithmetic-intensive tasks. Based on this end-to-end pipeline, we evaluate 8 LLMs spanning various model sizes and architectures.

**Remark.** Table 11 presents the results, where GPT-4 leads in overall performance. GPT-3.5 and Qwen-1.5-32B excel in FetaTab and NqText, respectively. The overall end-to-end scores demonstrate the challenges of our benchmark and also indicate considerable room for improvement in real-world document analysis for both RAG and LLMs.

Additionally, Table 12 presents the results when the LLMs respond to the questions with versus without RAG support. A significant decline in accuracy is observed when RAG is absent, underscoring

Table 12: Performance scores with and without RAG

| LLM and Strategy | FinHybrid | TatHybrid | PaperTab | PaperText | FetaTab | NqText |
|---|---|---|---|---|---|---|
| GPT-4 | **45.9** | **43.5** | **40.3** | **45.8** | **61.5** | **37.4** |
| GPT-4-NoRAG | 0.4 | 3.0 | 15.3 | 18.9 | 48.6 | 19.6 |
| Llama-3-8B | **37.9** | **22.5** | **35.5** | **42.3** | **56.6** | **31.7** |
| Llama-3-8B-NoRAG | 3.6 | 6.0 | 13.0 | 16.2 | 47.7 | 21.7 |

the LLMs' deficiency of related internal knowledge and their reliance on external documents. This reinforces the effectiveness of our benchmark for the evaluation and exploration of RAG approaches.

# 5 Conclusion

In this paper, we propose a novel benchmark to assess Retrieval Augmented Generation (RAG) methodologies in real-world document analysis scenarios. Our benchmark features diverse question types and encompasses thousands of unstructured documents with expert labels from financial and other domains. Meanwhile, we discuss interesting findings from our evaluations, covering data parsing, information retrieval, long context mechanism, generating strategies and end-to-end performance. We believe our benchmark will advance future research and production in unstructured document analysis.

# 6 Limitations

Despite the valuable contributions of this study, we acknowledge its limitations: (1) While the efficacy of parsing strategies was evaluated through the downstream Q&A performance, a direct comparison of the parsed content was not undertaken. This stems from the absence of well-defined standards for direct quality assessment in the scenario of document understanding. (2) This study did not extend to the in-depth analyses of noise sensitivity and hallucination. We will delve into these topics and conduct detailed analyses in our future work.

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

# A  Dataset Example

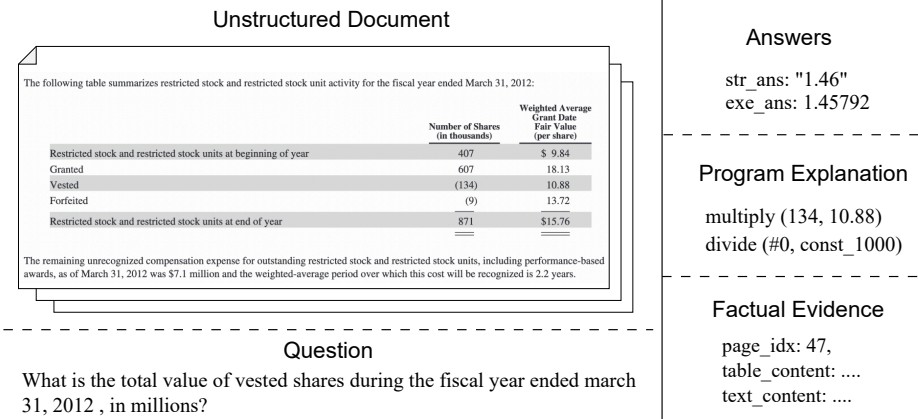

Figure 4: Data example of FinHybrid in UDA

Figure 4 presents a data item from the FinHybrid dataset, a prototypical representation common to all datasets within UDA. It comprises an original multi-page unstructured document containing tables and text, accompanied by related question-answer pairs. The datasets within UDA also feature additional explanations in different formats, such as the programmatic operation and factual evidence in FinHybrid.

# B  Experimental Details

## B.1  Prompt Templates for LLMs

Table 13 presents the prompt templates for LLM. We use the Chain-of-Thought approach for FinHybrid and TatHybrid datasets and also explore basic generation and Code-Interpreter strategies for FinHybrid, as detailed in Section 4.4. The example instruction of the prompt omits full context, just using a Q&A pair to guide LLM output toward the target answer format.

## B.2  Settings of Parsing Experiments

In our parsing experiments, we utilize 341 table-based questions from FinHybrid and 100 questions from PaperTab. FinHybrid provides cleanly extracted tables which we formatted into Markdown text as our well-parsed ground truth. To get the well-parsed content in PaperTab, we first use GPT-4-Omni to convert table images to Markdown, followed by manual corrections for inaccuracies. Since Markdown doesn't support the nested structure, we replicate the root content to represent hierarchical relationships.

## B.3  Settings of Indexing and Retrieval Experiments

In our indexing and retrieval experiments, we first extract raw text from documents, then segment it into 3000-character (about 500 words) chunks using the recursive-split method [47], ensuring a 10% overlap to mitigate information loss. Then the index is constructed on these chunks for the retrieval task. For evaluation, human-annotated factual evidence serves as the ground truth for retrieval. We measure evidence presence by calculating the ratio of the Longest Common Subsequence (LCS) length to the full evidence length.

## B.4  Long-context Policy

In our long-context experiments, we employ the Qwen-1.5-7B-32k and GPT4-Turbo-128k models, which can handle extended contexts but are still insufficient for quite long documents, such as financial reports exceeding 100k words (more than 150k tokens). For such cases, the strategy falls

Table 13: Prompt templates for LLMs in each dataset and task.

| Dataset | | Prompt Template |
|---|---|---|
| PaperTab and PaperText | System | You are a scientific researcher, given a section of an academic paper, please answer the question according to the context of the paper. The final answer output should be in the format of "The answer is: <answer>", and the <answer> should be concise with no explanation. |
| | User | ### Context: ... ### Question: Which Indian languages do they experiment with? ### Response: |
| | Assistant | The answer is: Hindi, English, Kannada, Telugu, Assamese, Bengali and Malayalam. |
| | User | ### Context: {context} ### Question: {question} ### Response: |
| FetaTab | System | Given a section of a document, plese answer the question according to the context. The final answer output should be in the format of "The answer is: <answer>", and the <answer> should be a natural sentence. |
| | User | ### Context: ... ### Question: When and in what play did Platt appear at the Music Box Theatre? with? ### Response: |
| | Assistant | The answer is: In 2016 and 2017, Platt played in Dear Evan Hansen on Broadway at the Music Box Theatre. |
| | User | ### Context: {context} ### Question: {question} ### Response: |
| NqText | System | Given a section of a document, plese answer the question according to the context. The final answer output should be in the format of "The answer is: <answer>", and the <answer> should be a paragraph from the context or a summarized short phrase. |
| | User | ### Context: ... ### Question: When will tour de france teams be announced? ### Response: |
| | Assistant | The answer is: 6 January 2018 |
| | User | ### Context: {context} ### Question: {question} ### Response: |
| FinHybrid and TatHybird (Chain-of-Thought) | System | You are a financial analyzer, given a section of a company's annual report, please answer the question according to the report context. Let's do this step by step. The final answer output should be in the format of "The answer is: <answer>", and the <answer> must be simple and short (e.g. just an accurate numerical value or phrases). |
| | User | ### Context: ... ### Question: What is the average price of the products? ### Response: |
| | Assistant | There are 8 products with a total price value of 1000 so the average value is 125.00. The answer is: 125.00 |
| | User | ### Context: {context} ### Question: {question} ### Response: |
| FinHybrid (Basic Generation) | System | You are a financial analyzer, given a section of a company's annual report, please answer the question according to the report context. The final answer output should be in the format of 'The answer is: <answer>', and the <answer> must be simple and short (e.g. just an accurate numerical value or phrases). |
| | User | ### Context: ... ### Question: What is the average price of the products? ### Response: |
| | Assistant | The answer is: 125.00 |
| | User | ### Context: {context} ### Question: {question} ### Response: |
| FinHybrid (Code Interpreter) | System | Given a section of a company's annual report and corresponding question, please generate the python codes to calculate the answer. You should firstly extract and list the relevant information from the context, and then write the arithmetical python codes in the following block format: ```python <python codes> ```If the answer does not require any calculation, you should directly write the answer in the format of "The answer is: <answer>". |
| | User | ### Context: ... ### Question: What is the average price of the products? ### Response: |
| | Assistant | The price of product 1,2,3,4 is 700, and the price of product 5,6,7,8 is 900. The python code is ```python   products = [700, 700, 700, 700, 900, 900, 900, 900] \n total_price = sum(products) \n average_price = total_price / len(products) \n print(average_price)``` |
| | User | ### Context: ... ### Question: What is the net income in 2009? ### Response: |
| | Assistant | The answer is: 1000 million |
| | User | ### Context: {context} ### Question: {question} ### Response: |

back to retrieving the top 30 most relevant data segments to serve as the input context. Due to the high cost of long context inference, we conduct this experiment on a subset of 600 documents, each coupled with a corresponding Q&A pair.

Table 14: Performance scores (Exact Match or LLM-Score) of LLMs using varying parsing strategies on table-based Q&A tasks (supplement to Table 5).

| Dataset | LLM Name | Well Parsed | GPT-4-Omni | Raw Text | CV | CV + LLM |
|---|---|---|---|---|---|---|
| Tabular FinHybrid (EM) | GPT-4-Turbo | 71.9 | **72.4** | 68.0 | 61.3 | 52.4 |
| | Llama-3-8B | **59.5** | 56.3 | 51.6 | 44.6 | 40.2 |
| PaperTab (LLM-Score) | GPT-4-Turbo | 54.3 | **56.3** | 54.3 | 53.5 | 52.5 |
| | Llama-3-8B | **46.8** | 41.3 | 40.3 | 36.0 | 37.0 |

Table 15: End-to-end answer scores using retrieved and human-annotated context (with the LLM evaluator, supplement to Table 7).

| LLM Name | Context Type | FinHybrid | TatHybrid | PaperTab | PaperText | FetaTab | NqText |
|---|---|---|---|---|---|---|---|
| Llama-3-8B | OpenAI Retrieval @5 | 37.9 | 22.5 | **39.9** | **43.2** | **69.0** | 82.5 |
| | Human-annotated | **51.0** | **35.9** | 39.1 | 41.2 | 68.0 | **85.2** |
| GPT-4-Turbo | OpenAI Retrieval @5 | 45.9 | 43.5 | 51.8 | 53.2 | **80.4** | 81.7 |
| | Human-annotated | **69.4** | **57.7** | **53.0** | **60.3** | 75.0 | **81.8** |

Table 16: Performance scores between long-context and RAG mechanism (with the LLM evaluator, supplement to Table 8).

| LLM Name | Input Type | FinHybrid | TatHybrid | PaperTab | PaperText | FetaTab | NqText |
|---|---|---|---|---|---|---|---|
| Qwen-1.5-7B | OpenAI Retrieval @5 | **21.0** | **26.6** | 40.0 | 44.2 | **68.0** | 73.0 |
| | Long Context | 3.0 | 20.9 | **40.8** | **48.0** | **68.0** | **77.3** |
| GPT-4-Turbo | OpenAI Retrieval @5 | **43.4** | **46.3** | **56.5** | 57.1 | **81.3** | 77.8 |
| | Long Context | 37.4 | 36.9 | 50.0 | **57.8** | 81.0 | **78.8** |

Table 17: End-to-end performance scores of different LLMs (with the LLM evaluator, supplement to Table 11).

| LLM Name | Avg | FinHybrid | TatHybrid | PaperTab | PaperText | FetaTab | NqText |
|---|---|---|---|---|---|---|---|
| GPT-4-turbo | **59.4** | **45.9** | **43.5** | **51.8** | **53.2** | **80.4** | 81.7 |
| GPT-3.5 | 55.3 | 36.6 | 33.9 | 46.2 | 51.1 | 76.1 | **87.9** |
| Llama-3-70B | 52.4 | 43.5 | 30.9 | 44.3 | 43.3 | 72.4 | 80.3 |
| Mixtral-8x7B-v0.1 | 50.0 | 28.4 | 22.5 | 48.2 | 50.7 | 67.9 | 82.5 |
| Llama-3-8B | 49.2 | 37.9 | 22.5 | 39.9 | 43.2 | 69.0 | 82.5 |
| Qwen-1.5-32B | 45.8 | 31.3 | 27.9 | 39.7 | 43.5 | 66.5 | 65.6 |
| Mistral-7B-v0.2 | 45.1 | 18.2 | 15.9 | 41.8 | 45.3 | 66.3 | 83.0 |
| Qwen-1.5-7B | 43.6 | 17.0 | 22.6 | 39.5 | 44.9 | 66.1 | 71.7 |

## B.5 LLM Evaluator

We also incorporate the LLM-based method for a more comprehensive evaluation. Following the previous work [32], we implement our LLM-based evaluator with few-shot in-context learning. Given the question, ground-truth answer, and the generated response, the LLM evaluator scores the response from 0 to 4 based on correctness. For the FinHybrid and TatHybrid datasets, where the answers are digits or extractive words, the existing numeric or word-level matching is sufficient for scoring. For the remaining datasets, PaperTab, PaperText, FetaTab, and NqText, with natural-language answers, we apply the LLM evaluator to re-evaluate all of our experiments. The results are illustrated from Table 14 to Table 17, where the scores are normalized to the 100-scale for clarity.

Our findings are further validated by the results from the LLM evaluator. For example, the long-context and RAG mechanism yield comparable results on free-form and knowledge-based tasks, and GPT-4-omni and raw-text parsing methods demonstrate decent performance. Unlike the rule-based evaluation, the LLM evaluator works for varied but similar expressions and produces generally higher absolute scores, while it may decrease the interpretability of how the scores are determined.

