# UDA

UDA (Unstructured Document Analysis) is a benchmark suite for Retrieval Augmented Generation (RAG) in real-world document analysis. Each entry in the UDA dataset is organized as a *document-question-answer* triplet, where a question is raised from the document, accompanied by a corresponding ground-truth answer. The documents are retained in their original file formats without parsing or segmentation; they consist of both textual and tabular data, reflecting the complex nature of real-world analytical scenarios.

The UDA dataset comprises six subsets spanning finance, academia, and knowledge bases, encompassing 2965 documents and 29590 expert-annotated Q&A pairs. To build this dataset, we first collect the Q&A labels from the open-released datasets (i.e., source datasets). Then we conduct a series of essential constructing actions, including source-document identification, categorization, data cleaning and data standardization.

## Links

**Dataset Link:** https://huggingface.co/datasets/qinchuanhui/UDA-QA

**Croissant MetaData URL:** https://huggingface.co/api/datasets/qinchuanhui/UDA-QA/croissant

**Benchmark Repository** (with codes): https://github.com/qinchuanhui/UDA-Benchmark (also have Croissant metadata)

We also include a persistent dereferenceable identifier, i.e., a **DOI**, for our UDA dataset: 10.57967/hf/2536

# 1 Authorship

## 1.1 Publishers and Owners

### Publishing Organization(s)

Tsinghua University, National University of Singapore, Shanghai Qi Zhi Institute

### Industry Type(s)

Academic - Tech

### Contact Detail(s)

- Publishing POC: Yulong Hui
- Affiliation: Tsinghua University, Shanghai Qi Zhi Institute
- Contact: qinchuanhui@gmail.com

### Author(s)

- Yulong Hui, Tsinghua University / Shanghai Qi Zhi Institute
- Yao Lu, National University of Singapore
- Huanchen Zhang,  Tsinghua University / Shanghai Qi Zhi Institute

## 1.2 Funding Sources

**Institution(s)**

- Tsinghua University
- Shanghai Qi Zhi Institute

# 2. Dataset Description

## 2.1 Dataset Overview

**Data Subject(s)**

- Data about common world knowledge
- Data about public academic papers
- Data about public financial reports

**Content Description**

UDA dataset includes six sub-datasets across three pivotal domains: finance, academia, and knowledge bases. Each data entry is organized as a *document-question-answer* triplet, where a question is raised from the document, accompanied by a corresponding ground-truth answer. The documents are retained in their original file formats without parsing or segmentation; they consist of both textual and tabular data, reflecting the complex nature of real-world analytical scenarios.

**Dataset Snapshot**

| Category | Data |
|---|---|
| Size of Dataset | 6.7GB |
| Number of sub-dataset | 6 |
| Number of Q&A Instances | 29590 |
| Number of Documents | 2965 |
| Number of Fields | 4 or more |
| Average Q&A Instances Per Document | 9.98 |

**Descriptive Statistics**

| Sub Dataset (Source Domain) | Doc Format | Doc Num | Q&A Num | Avg #Words | Avg #Pages | Total Size | Q&A Types |
|---|---|---|---|---|---|---|---|
| FinHybrid (Finance) | PDF | 788 | 8190 | 76.6k | 147.8 | 2.61 GB | arithmetic |
| TatHybrid (Finance) | PDF | 170 | 14703 | 77.5k | 148.5 | 0.58 GB | extractive, counting, arithmetic |
| PaperTab (Academia) | PDF | 307 | 393 | 6.1k | 11.0 | 0.22 GB | extractive, yes/no, free-form |
| PaperText (Academia) | PDF | 1087 | 2804 | 5.9k | 10.6 | 0.87 GB | extractive, yes/no, free-form |
| FetaTab (Wikipedia) | PDF & HTML | 878 | 1023 | 6.0k | 14.9 | 0.92 GB | free-form |
| NqText (Wikipedia) | PDF & HTML | 645 | 2477 | 6.1k | 14.9 | 0.68 GB | extractive |

**Above Table:** An overview of sub-datasets in UDA and their statistics.

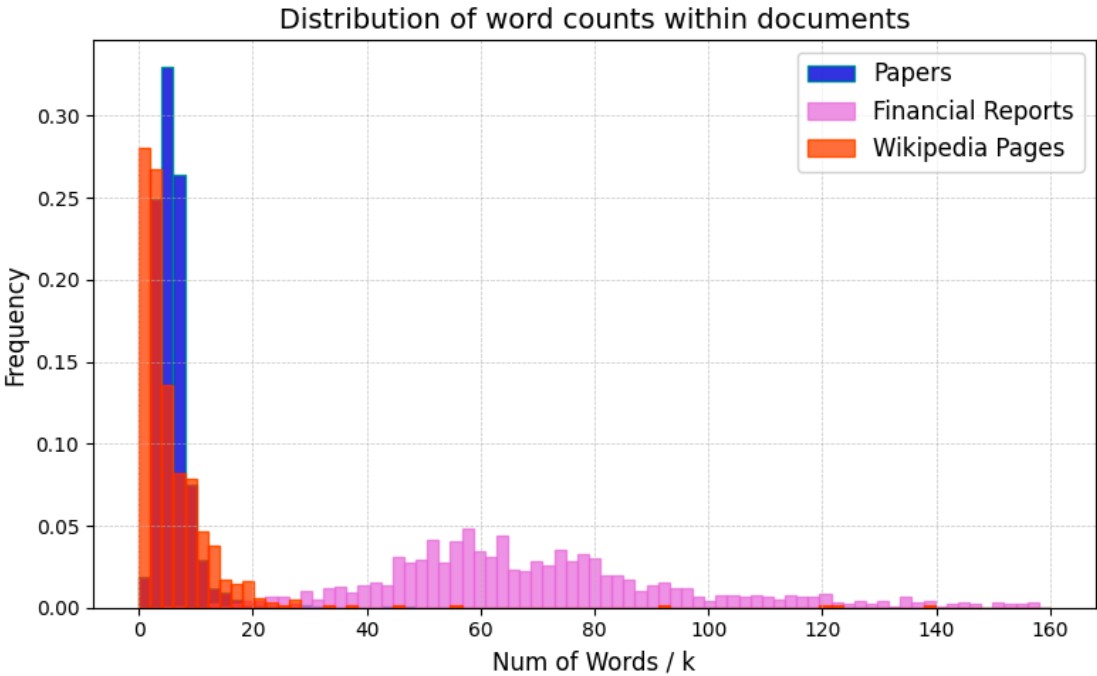

**Above Figure:** We present the distribution of word counts within documents across the three involved source domains.

## 2.2 Example of Data Points

### Primary Data Modality

- Unstructured document data
- Text Data
- Tabular Data

## Data Format(s)

- PDF or HTML documents
- CSV or Parquet tables for Q&A labels
- JSON files for more detailed Q&A information

## Data Fields

| Field Name | Field Value | Description | Example |
|---|---|---|---|
| `doc_name` | string | name of the source document file | 1912.01214 |
| `q_uid` | string | unique id of the question | 9a05a5f4351db75da371f7ac12eb0b03607c4b87 |
| `question` | string | raised question | Which datasets did they experiment with? |
| `answer`
or `answer_1`, `answer_2`
or `short_answer`, `long_answer` | string | ground truth answer | Europarl, MultiUN |

**Additional Notes:** Some sub-datasets may have multiple ground_truth answers, where the answers are organized as `answer_1`, `answer_2` (in FinHybrid, PaperTab and PaperText) or `short_answer`, `long_answer` (in NqText); In TatHybrid, the answer is organized as a sequence, due to the involvement of the multi-span Q&A type. Additionally, some sub-datasets may have unique data fields. For example, `doc_url` in FetaTab and NqText describes the Wikipedia URL page, while `answer_type` and `answer_scale` in TatHybrid provide extended answer references.

For more details and the data preview, please refer to https://huggingface.co/datasets/qinchuanhui/UDA-QA. And the source document files are placed in a separate directory, accessible through the `doc_name` field.

## Typical Data Point

```
# FinHybrid
{ 'doc_name': 'ADI_2009',
  'q_uid': 'ADI/2009/page_59.pdf-2',
  'question': 'What is the expected growth rate in amortization expense in 2010?',
  'answer_1': '-27.0%',
  'answer_2': ' -0.26689'}

# TatHybrid
{ 'doc_name': 'lifeway-foods-inc_2019',
  'q_uid': 'd3ef8fd4b3583394a5552b6bf7d0a292',
  'question': 'What is the total cost of goods sold for years 2018 and 2019
respectively?',
  'answer':   [ "$ 77,492", "$ 71,513" ],
  'answer_scale': 'thousand',
  'answer_type': 'multi-span'
}

# PaperTab and PaperText
{ 'doc_name': '1904.10503',
  'q_uid': '5a65ad10ff954d0f27bb3ccd9027e3d8f7f6bb76',
```

```
    'question': 'Which other approaches do they compare their model with?',
    'answer_1': 'Akbik et al. (2018), Link et al. (2012)',
    'answer_2': ' They compare to Akbik et al. (2018) and Link et al. (2012).',
    'answer_3': ' ',
}

# FetaTab
{ 'doc_name': 'Ben Platt (actor)',
  'q_uid': '14717',
  'question': ' When and in what play did Platt appear at the Music Box Theatre?',
  'answer': 'In 2016 and 2017, Platt played in Dear Evan Hansen on Broadway at the
Music Box Theatre.',
  'doc_url': 'http://en.wikipedia.org/wiki/Ben_Platt_(actor)',
}

# NqText
{ 'doc_name': 'Supreme Court of the United States',
  'q_uid': '-8377155967940204000',
  'question': 'who determines the size of the supreme court?',
  'short_answer': 'Congress added',
  'long_answer': "Article III of the United States Constitution does not specify the
number of justices. The Judiciary Act of 1789 called for the appointment of six ``
judges ''. Although an 1801 act would have reduced the size of the court to five
members upon its next vacancy, an 1802 act promptly negated the 1801 act, legally
restoring the court's size to six members before any such vacancy occurred. As the
nation's boundaries grew, Congress added justices to correspond with the growing number
of judicial circuits: seven in 1807, nine in 1837, and ten in 1863.",
  'doc_url':'https://en.wikipedia.org//w/index.php?
title=Supreme_Court_of_the_United_States&oldid=815925358'
}
```

# 3  Motivations & Intentions

## 3.1  Motivations

**Purpose(s)**

- Research

**Domain(s) of Application**

Retrieval Augmented Generation, Document Analysis, Natural Language Processing, Large Language Model

**Motivating Factor(s)**

Provide a benchmark suite for Retrieval Augmented Generation (RAG) in real-world document analysis.

## 3.2 Intended Use

### Dataset Use(s)

Safe for research use

### Direct Use Case(s)

Question-answering tasks on real-world unstructured documents

### Extended Use Space(s)

- Evaluate the end-to-end performance of RAG Q&A pipelines.
- Evaluate the effectiveness of retrieval strategies using the provided evidence (in `extended_qa_info` json files).
- Directly assess the performance of LLMs in numerical reasoning and table reasoning.
- Assess the effectiveness of parsing strategies on unstructured PDF documents.

## 3.3 Use in ML or AI Systems

### Dataset Use(s)

- Training
- Testing
- Validation
- Development or Production Use
- Fine Tuning

We utilize the UDA dataset for evaluation and benchmarking, while its representation of *document-question-answer* pairs can serve many purposes. For example, users can leverage it to train the retrieval model for extensive contexts, fine tune LLMs for specific domains and unstructured data patterns, evaluate the performance of diverse modules, and develop end-to-end analytical frameworks.

### Usage Guideline(s)

Please check https://github.com/qinchuanhui/UDA-Benchmark for usage guidelines.

# 4 Dataset Construction

## 4.1 Label Collection

We collect the Q&A labels from the open-released datasets (i.e., source datasets), which are all annotated by human participants.

### Methodology Detail(s)

**Source:** Open-released Datasets

**Collection Cadence:** Static: Data was collected once from single or multiple sources.

**Is this source considered sensitive or high-risk?** [No]

### Source Description(s)

- **Source: FINQA:** https://github.com/czyssrs/FinQA/tree/main
  [1] CHEN, Z., CHEN, W., SMILEY, C., SHAH, S., BOROVA, I., LANGDON, D., MOUSSA, R., BEANE, M., HUANG, T.-H., ROUTLEDGE, B., ET AL. Finqa: A dataset of numerical reasoning over financial data. arXiv preprint arXiv:2109.00122 (2021).

- **Source: TAT-DQA:** https://nextplusplus.github.io/TAT-DQA/
  [2] ZHU, F., LEI, W., FENG, F., WANG, C., ZHANG, H., AND CHUA, T.-S. Towards complex document understanding by discrete reasoning. In Proceedings of the 30th ACM International Conference on Multimedia (2022), pp. 4857–4866.

- **Source: Qasper:** https://allenai.org/data/qasper
  [3] DASIGI, P., LO, K., BELTAGY, I., COHAN, A., SMITH, N. A., AND GARDNER, M. A dataset of information-seeking questions and answers anchored in research papers. arXiv preprint arXiv:2105.03011 (2021).

- **Source: FetaQA:** https://github.com/Yale-LILY/FeTaQA
  [4] NAN, L., HSIEH, C., MAO, Z., LIN, X. V., VERMA, N., ZHANG, R., KRYS´ CIN´ SKI, W., SCHOELKOPF, H., KONG, R., TANG, X., ET AL. Fetaqa: Free-form table question answering. Transactions of the Association for Computational Linguistics 10 (2022), 35–49.

- **Source: NaturalQuestions:** https://ai.google.com/research/NaturalQuestions
  [5] KWIATKOWSKI, T., PALOMAKI, J., REDFIELD, O., COLLINS, M., PARIKH, A., ALBERTI, C., EPSTEIN, D., POLOSUKHIN, I., DEVLIN, J., LEE, K., ET AL. Natural questions: a benchmark for question answering research. Transactions of the Association for Computational Linguistics 7 (2019), 453–466.

## 4.2 Source-Document Identification

The integrity of original documents is crucial for the fidelity of document analysis. However, most of the source datasets only offer well-parsed and segmented partial content without the complete document. To address this problem, we perform a comprehensive source-document identification process.

### Methodology Detail(s)

- Search for the publicly available source document according to the metadata from the above source datasets.

- Verify and collect the documents that exactly match the content fragments. We employ meticulous manual inspection, paired with the utilization of PyPDF for automated data parsing and comparison.

- Exclude inaccessible and unavailable documents, such as damaged files, withdrawn papers, and unauthorized files.

### Document Source Platform(s)

- Public earning reports: https://www.sec.gov/edgar/
- Public company annual reports: https://www.annualreports.com/
- Public papers: https://arxiv.org/
- Wikipedia knowledge base:  https://www.wikipedia.org/

Our methodology aligns with the originating documents and platforms of the source datasets, ensuring no introduction of sensitive or high-risk new elements.

## 4.3  Data Transformation

### Transformation(s) Applied

- Cleaning missing values
- Categorizing queries
- Converting data types
- Unifying data formats

### Methodology Detail(s)

**Cleaning missing values:** The human-generated questions may be unanswerable. Thus, we remove the Q&A items that lack available answers. Additionally, documents lacking any valid Q&A pairs are also removed.

**Categorizing queries:** Categorize the queries from Qasper into table-based and text-based, according to the human-annotated evidence. This forms the PaperTab and PaperText sub-datasets, which serve to facilitate applications on different data patterns.

**Converting data types:** In the NaturalQuestions dataset, answers are encapsulated within HTML tokens. We convert these tokens into natural language, thereby enhancing interpretability and standardizing metrics for the evaluation.

**Unifying data formats:** Data representations across source datasets are inherently heterogeneous, displaying a broad spectrum of structural paradigms. To address this, we reconstitute the various data forms into structured tables (detailed in Section 2.2).

# 5  License and Maintenance

## 5.1  License

### Licenses of Source Datasets

| Dataset Name | License |
| --- | --- |
| FINQA | MIT license |
| TAT-DQA | CC BY 4.0 |
| Qapser | CC BY 4.0 |
| FetaQA | CC BY-SA 4.0 |
| NaturalQuestions | CC BY-SA 3.0 |

### License of UDA Dataset

`License CC BY-SA 4.0`

In adherence to the licensing protocols of the above source datasets, our UDA dataset is distributed under the Creative Commons Attribution-ShareAlike 4.0 International (CC-BY-SA 4.0) License.

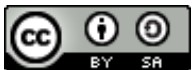

## 5.2  Maintenance

The dataset and benchmarking suite are openly hosted on both the HuggingFace and GitHub platforms, accessible via the following URL:

- https://huggingface.co/datasets/qinchuanhui/UDA-QA

- https://github.com/qinchuanhui/UDA-Benchmark.

These repositories ensure sustained, reliable maintenance and provide straightforward open access to the resources.

We invite the community to engage with us by providing updates and contributions, and reporting issues directly via these platforms. Our team is committed to actively incorporating new feedback to continually enhance the dataset's utility. Moreover, users are encouraged to expand, augment, and build upon the UDA benchmark, in accordance with the terms of the Creative Commons Attribution-ShareAlike 4.0 International (CC-BY-SA 4.0) License.

# 6  Ethics, Responsibilities

## 6.1  Ethical Considerations

**Consent and Copyright:** Our dataset is based on the source datasets that have been openly released, adhering to their respective licensing agreements as outlined in section 5.1.  We perform construction upon these datasets within the boundaries of their usage protocols.

**Security and Personal Privacy :** Our dataset focuses on publicly available financial reports, papers and wikipedia-pages, which have been widely accepted in prior open-released datasets. UDA dataset does not introduce any security-sensitive information or items that would compromise personal privacy.

**Sensitive/Offensive Content and Discrimination:** Our dataset is free from sensitive/offensive content; it does not include data about racial or ethnic origins, sexual orientations, religious beliefs or political opinions. The documents included are publicly available and have been commonly utilized and accepted by the broader community.

**Potential Negative Impacts:** Our dataset emphasizes the positive development in document analysis techniques, while we should also consider the potential risks, such as the propagation of misinformation and inappropriate model shaping. Thus, we advocate for the responsible applications of our dataset within the sphere of research and education.

## 6.2 Responsibilities

The UDA benchmark suite is designed exclusively for academic and educational purposes. Diligent efforts have been made by the authors to ensure security and adhere to ethical standards.

The authors bear all responsibility in case of any violation of rights or any other legal issues that arise from the use of this dataset. The datasets within UDA are distributed under the Creative Commons Attribution-ShareAlike 4.0 International License (CC-BY-SA 4.0). Users engaging with these datasets must agree to comply with the terms of this license.

## 7. Reproducibility

To support the reproducibility of our benchmarking experiments, all requisite codes, datasets, and instructions are accessible via the UDA GitHub repository at https://github.com/qinchuanhui/UDA-Benchmark. Our methodology aligns with the officially recommended ML reproducibility checklist, and we ensure that all results are easily reproducible. Detailed guidelines for use are included within the repository, and we encourage researchers to utilize these resources to replicate our work, as well as to pursue additional enhancements and explorations.