# OpenReview forum: "UDA: A Benchmark Suite for Retrieval Augmented Generation in Real-World Document Analysis"
_NeurIPS.cc/2024/Datasets_and_Benchmarks_Track — NeurIPS 2024 Track Datasets and Benchmarks Poster_

### Official Review · Reviewer_ubM5 · 2024-06-25
**UDA: A Benchmark Suite for Retrieval Augmented Generation in Real-World Document Analysis**

**Rating:** 8
**Confidence:** 4
**Correctness:** yes
**Clarity:** yes

**Review:**

Please refer to Strengths and Limitations

**Strengths:**

Comprehensive Dataset: UDA includes a diverse set of documents and Q&A pairs, covering multiple domains and various types of queries, which enhances its applicability to real-world scenarios.

Detailed Evaluation: The benchmark provides a thorough evaluation of different RAG components, including parsing methods, retrieval strategies, and generation techniques. This allows for a granular analysis of each component's impact on overall performance.

Advanced Techniques: The study explores the use of advanced techniques like Chain-of-Thought (CoT) reasoning and Code Interpreters, highlighting their potential benefits in improving answer quality, especially for complex queries.

**Additional Feedback:**

no

**Documentation:**

yes

**Limitations:**

Lacks analysis of bad cases, such as noise sensitivity and hallucination issues.

When the recall of retrieval segments is accurate, how is the generation performance of the large language model (LLM)?

**Opportunities For Improvement:**

Although the datasets in this paper are sourced from other datasets, it provides very comprehensive experiments, which are very valuable. In addition to automatic metrics, it is also recommended to include necessary human evaluation. Although metrics like span-level F1-score have been validated, they do not fully indicate that the results on this dataset will closely align with human evaluation.

**Relation To Prior Work:**

yes

**Summary And Contributions:**

The article introduces the Unstructured Document Analysis (UDA) benchmark suite, designed to evaluate the performance of Retrieval-Augmented Generation (RAG) systems in handling real-world unstructured documents. The UDA dataset includes 2,965 documents and 29,590 expert-annotated Q&A pairs from domains such as finance, academia, and general knowledge. The benchmark aims to assess the effectiveness of various components in RAG workflows, including data parsing, indexing, retrieval, and question answering strategies.

---

> ### Author Rebuttal · Authors · 2024-08-16
>
> Thank you for your supportive and insightful comments.
>
> **Human evaluation**
>
> We appreciate your suggestion on human evaluation and acknowledge its importance. To bolster our evaluation's robustness and better align with human judgment, we've integrated LLM-based evaluation (as detailed in our response to reviewer FaEw). Given the scope of our benchmark and the challenges associated with human evaluation, we have yet to implement it extensively but will consider incorporating it in future research.
>
> **Lacks analysis of bad cases, such as noise sensitivity and hallucination.**
>
> Thanks for your valuable suggestion. While we provide limited case analysis in Table 10,  we will delve into these topics and conduct detailed analyses in our future work.
>
> **When the recall of retrieval segments is accurate, how is the generation performance.**
>
> We acknowledge the need for clarity and will improve the presentation of related results in Table 7. This table showcases the generation performance with human-annotated contexts where the provided segment accurately contains the precise evidence. It also shows the performance on OpenAI-based contexts, where the top-5 related segments are supplied based on the embedding model's assessments. The results indicate that tasks requiring detailed evidence for accurate arithmetic reasoning, such as FinHybrid and TatHybrid, can benefit substantially from the accurate segment. Additionally, for knowledge-based questions, incorporating multiple relevant segments (top-5 retrieval) can yield comparable results with a single precise segment. Moreover, even when the retrieval segment is accurate (human-annotated), there remains a clear opportunity for LLMs to enhance the quality of generation, underscoring the challenges posed by our benchmark.

---

### Official Review · Reviewer_zcFM · 2024-07-24
**Reviews**

**Rating:** 6
**Confidence:** 4

**Review:**

Please refer to the Strengths and Limitations.

**Strengths:**

1. The benchmark collects the corresponding original format questions for each Q&A pair, allowing for the evaluation of the impact of different real-world document parsing methods on RAG (retrieval-augmented generation).
2.  The dataset includes multiple different scenarios and various forms of data, enabling it to closely align with real-world applications.
3.  The experiments are thorough, exploring the analytical results that the dataset can provide at different stages of RAG, including segmentation, retrieval, and reasoning.

**Additional Feedback:**

n/a

**Clarity:**

The paper is well-structured, with clear ideas, and the experiments and analysis are carried out in a hierarchical manner. It is quite well written.

**Correctness:**

The evaluation methods and experiment design are appropriate and performed correctly.

**Documentation:**

The data collection process is generally clear. However, as mentioned in the limitations section, the specific steps for collecting documents that lack the complete document are not clearly described.

**Ethics:**

The data is collected and organized from publicly available datasets, so I have no ethical concerns with the submission.

**Limitations:**

1. As shown in Table 7, using human-annotated context does not improve the response results for PaperTab and PaperText. This suggests that such questions do not seem conducive to the exploration of RAG (retrieval-augmented generation), and this part of the data does not contribute to the evaluation of retrieval models. The paper does not explain why this data still needs to be included in the benchmark.
2. The significance of bolded data in the tables is not explained in the paper. It appears that not only the best results are bolded, as some of the best results are not highlighted.
3. In line 117, the paper mentions that for parts of the data that do not provide the complete document, the authors performed a comprehensive identification process. This section lacks details on how the search was conducted and does not explain how the accuracy of the retrieval was evaluated.
4. It seems that results without RAG are missing for comparison, which is necessary to validate the effectiveness of RAG data. Additionally, it is important to rule out the possibility that large language models (LLMs) have already seen these data during pre-training and have retained related knowledge.

**Opportunities For Improvement:**

Please refer to the limitations.

**Relation To Prior Work:**

This benchmark features real-world RAG (retrieval-augmented generation) Q&A data in various forms, including text and tables, across multiple domains. It also provides knowledge sources at multiple granularities, distinguishing it clearly from previous benchmarks.

**Summary And Contributions:**

This paper proposes a benchmark utilizing real-world documents for retrieval-augmented tasks. The study collects relevant Q&A pairs from multiple datasets, including FinHybrid, TatHybrid, PaperTab, PaperText, FetaTab, and NqText, and extracts the original texts associated with each Q&A pair for retrieval purposes. By analyzing various parsing methods, retrieval strategies, comparisons with retrieval-augmented generation (RAG) and long-context approaches, and techniques such as Chain-of-Thought reasoning, the study explores the accuracy improvements in responses to real-world questions through the retrieval of unstructured data. Experimental analysis demonstrates that this dataset effectively supports retrieval-augmented generation tasks using real-world documents.

---

> ### Author Rebuttal · Authors · 2024-08-16
>
> Thank you for your positive and valuable comments.
>
> **Add results without RAG**
>
> Thanks for the insightful suggestion, we have just conducted Q&A tasks using LLMs without RAG (with results shown in the following table). For the generation with RAG, we keep the same policy from the paper,  using the OpenAI embedding model to retrieve top-5 related chunks.
>
> |            |             | FinHybrid | TatHrbrid | PaperTab | PaperText | FetaTab | NqText |
> | ---------- | ----------- | --------- | --------- | -------- | --------- | ------- | ------ |
> | GPT-4      | without RAG | 0.4       | 3.0       | 15.3     | 18.9      | 48.6    | 19.6   |
> | GPT-4      | with RAG    | 45.9      | 43.5      | 40.3     | 45.8      | 61.5    | 37.4   |
> | Llama-3-8B | without RAG | 3.6       | 6.0       | 13.0     | 16.2      | 47.7    | 21.7   |
> | Llama-3-8B | with RAG    | 37.9      | 22.5      | 35.5     | 42.3      | 56.6    | 31.7   |
>
> We can observe a significant decline in LLM performance without RAG and external document context, which illustrates the effectiveness of our benchmark on the exploration of RAG approaches. Specifically, without RAG, for the queries in FinHybrid and TatHybrid, which are based on exact financial reports,  LLMs may fail to respond due to the lack of related internal knowledge. Performance on PaperTab and PaperText also suffered significantly, because the queries and answers are derived from the specific paper details. While LLMs retain some general knowledge for Wiki-based FetaTab and NqText, the improvement with RAG is still evident.
>
> **Explanation of experiments and datasets**
>
> > As shown in Table 7, using human-annotated context does not improve the response results for PaperTab and PaperText ......  why this data still needs to be included in the benchmark.
>
> Thanks for your thoughtful comment. We would like to clarify that Table 7 compares the effects between employing a single human-annotated chunk and five model-auto-retrieved context chunks, showing the performance of the retrieval strategy. Results indicate that the chunks retrieved by the OpenAI embedding model may contain the accurate context or provide other relevant data aiding the LLM generation. In some cases, the final accuracy metrics remain similar, demonstrating that subjective human annotations can be automated by well-designed chunking and retrieval mechanisms.
>
> The effectiveness of our datasets is also evidenced by the marked decrease in LLM performance without RAG, as shown in the above experiments. This suggests the need for well-organized external contexts to improve generation, highlighting the utility of these data to explore retrieval approaches. Additionally, the inclusion of our sub-datasets allows for comprehensive evaluations across various domains and query types, facilitating analyses on diverse aspects, such as parsing strategies, long-context mechanisms and LLM performance.
>
>
>
> **Bolded data in the tables is not explained**
>
> Thanks for pointing out this, and we will clarify the illustration in the next version of our paper.
>
>
>
> **Details on document identification**
>
> > In the comprehensive identification process, how the search was conducted and how the accuracy of the retrieval was evaluated.
>
> Thanks for the advice. In the next version of our paper, we will elaborate on our source document identification process, which includes retrieval (search), verification, and cleaning. (1) Retrieval: the original documents source from certain platforms, including Wikipedia, arXiv, the SEC website, and the AnnualReports website. We conduct retrieval on these platforms using metadata from existing datasets, such as titles, timestamps, and document types. (2) Verification: we verify the accuracy of the retrieved document by cross-referencing the content fragments in existing datasets, collecting only exact matches. Specifically, we employ the PyPDF library for automatic data parsing and comparison, supplemented by manual inspection where automatic methods are insufficient. (3) Cleaning: we remove the documents that are inaccessible or not available, such as damaged files, withdrawn papers, and unauthorized files.

---

> > ### Comment · Reviewer_zcFM · 2024-08-27
> > **Thank you for the responses**
> >
> > Thank you for the detailed responses. I tend to keep my initial rating.

---

### Official Review · Reviewer_FaEw · 2024-07-25
**Offical Review**

**Rating:** 6
**Confidence:** 4
**Clarity:** Yes

**Review:**

Quality: The paper demonstrates high quality in its comprehensive approach to benchmarking RAG systems for document analysis. The authors have clearly put significant effort into creating a diverse and challenging dataset, and their evaluation methodology is thorough and well-structured.

Clarity: The paper is generally well-written and organized logically. The authors clearly explain their methodology, experiments, and findings. However, some technical details could be further elaborated for better reproducibility.

Originality: The UDA benchmark presents a novel contribution to the field. While there are existing Q&A datasets, UDA's focus on unstructured documents across multiple domains, including financial and academic papers, sets it apart. The comprehensive evaluation of various RAG components adds to its originality.

Significance: This work is significant as it addresses a crucial gap in evaluating RAG systems for real-world document analysis tasks. The benchmark could become a valuable tool for researchers and practitioners in developing more robust and effective document analysis systems.

Pros:
- Addresses a real-world need for evaluating document analysis systems on unstructured data covering multiple domains and question types
- They retain the original document formats which mirrors real-world challenges
- The authors compare RAG methods with long-context language models and also evaluate multiple state-of-the-art language models



Cons:
- The evaluation metrics are almost rule-based methods. Did you try LLM-based evaluators?
- The ground truth answers in the dataset are all taken from the source datasets. However, these answers are not lilely long-formed as current LLMs tend to generate.

**Strengths:**

The main strengh of this paper is the benchmark dataset they curated. In addition, with the benchmark dataset, the authors conducted comprehensive experiments and the observations are insightful.

**Additional Feedback:**

No

**Correctness:**

The dataset is curated by a combination of public datasets. The evaluation methods in this paper has clear limitations.

**Documentation:**

Yes

**Ethics:**

No ethical concerns

**Limitations:**

This paper lacks of a limitation discussion. See Opportunities For Improvement.

**Opportunities For Improvement:**

- Better evaluation methods such as LLM-based evaluations
- Analysis of the dataset to prove the quality of the benchmark

**Relation To Prior Work:**

Yes

**Summary And Contributions:**

This paper introduces UDA (Unstructured Document Analysis), a new benchmark suite for evaluating retrieval-augmented generation (RAG) methods on real-world document analysis tasks. The key contributions of the paper are:

- A benchmark dataset: UDA comprises 2,965 real-world documents and 29,590 expert-annotated Q&A pairs across finance, academia, and general knowledge domains. It includes diverse question types and retains documents in their original unstructured formats.
- Comprehensive evaluation: The authors evaluate various components of the RAG pipeline, including data parsing, indexing and retrieval, and generation strategies. They also compare RAG approaches with long-context language models.


They also conclude with some key findings:

- Data parsing remains challenging, with simple raw text extraction often performing comparably to more complex computer vision-based methods.
- Retrieval quality significantly impacts performance, especially for arithmetic reasoning tasks.
- RAG outperforms long-context models on numerical reasoning tasks.
- Chain-of-Thought prompting improves performance for arithmetic tasks.

---

> ### Author Rebuttal · Authors · 2024-08-16
>
> Thank you for your positive and valuable comments.
>
> **Try LLM-based evaluators**
>
> We appreciate the suggestion and have incorporated the LLM-based method for a more comprehensive evaluation. Following the work in [1], we implement our LLM-based evaluator with few-shot in-context learning. Given the question, ground-truth answer, and the generated response, the LLM evaluator scores the response on a 0-4 scale based on correctness. For the FinHybrid and TatHybrid datasets, where the answers are digits or extractive words, the existing numeric or word-level matching is sufficient for scoring. For the remaining datasets—PaperTab, PaperText, FetaTab, and NqText, with natural-language answers, we apply the LLM evaluator to re-evaluate all of our experiments. The results are illustrated in the attached PDF, where the scores are normalized to the 100-scale for clarity.
>
> Our findings are validated by the results from the LLM evaluator. For example, the long-context and RAG mechanism yield comparable results on free-form and knowledge-based tasks, and GPT-4-omni and raw-text parsing methods produce decent performance. Unlike rule-based evaluation, the LLM evaluator works for varied but similar expressions and produces generally higher absolute scores, while it may decrease the interpretability of how the scores are determined.
>
>
>
> **Explanation of data form**
>
> > These answers are not likely long-formed as current LLMs tend to generate.  Analysis of the dataset to prove the quality of the benchmark
>
> Thanks for your thoughtful comment, and we want to clarify that our study concentrates on document-specific Q&A tasks. Unlike the open-domain Q&A or creative content generation tasks with long-form answers, our task necessitates the answers directly extracted or inferred from the document, requiring a few digits or sentences. These concise answers can be aligned with current LLMs by formatting prompts shown in Appendix B.1. Furthermore, while the final answers are not lengthy, the generation may benefit from the long-form capabilities of LLMs, such as elaborating the intermediate reasoning steps to enhance final output, as outlined in Section 4.4's Chain-of-Thought strategy.
>
> Additionally, as detailed in Section 3 and the supplementary material, our dataset demonstrates extensive coverage, diversity, and generality across multiple domains and varied query types, such as boolean, extractive, counting, free-form, and arithmetic. It leverages a large amount of documents and Q&A pairs with a broad spectrum of word counts. Besides, our benchmark evaluates multiple RAG modules, and the results reveal that the benchmark is challenging for existing solutions.
>
>
>
> **Technical details**
>
> > The authors clearly explain their methodology, experiments, and findings. However, some technical details could be further elaborated for better reproducibility.
>
> Thanks for your valuable comment. We made our entire dataset, benchmark tools, experimental codes, and code-usage instructions available on the GitHub repository (referenced in the Abstract), bolstering the reproducibility of our study. We will also add more technical details in the next version of our paper, such as the implementation of our tools and the methodologies of the experimental codes.
>
>
>
> **Add an explicit limitation section**
>
> Thanks for the advice. We will add an explicit and detailed limitation section in the next version of our paper. The major limitation of our work can be summarized as: due to the evaluation and annotations that may be non-standard or non-scalable, some intricate analyses remain in future work, including direct comparison of parsed content, and in-depth analysis of noise sensitivity and hallucination.
>
>
>
> [1] Liu et al., Calibrating LLM-Based Evaluator. (LREC-COLING 2024)

---

### Official Review · Reviewer_snvC · 2024-07-27
**Useful benchmark for parsing documents and RAG**

**Rating:** 6
**Confidence:** 3
**Correctness:** Yes.

**Review:**

Pros:
- The paper did decent work on integrating original documents with the collected QA pairs.
- This work fits community’s interest by going beyond clean text analysis to real-world unstructured document analysis.
 - The paper is well writing.

Cons:
- Regarding data parsing evaluation, the author use the downstream QA metrics to compare different parsing techniques. This is one way for comparison. However, it would be also essential to directly compare the parsed content with the ground truth well-parsed content, which is a more explicit.

**Strengths:**

- The work explicitly covers the document parsing step in the pipeline, and benchmarks several sota parsing techniques, which reveals great insight for documents parsing and draw the community attention to solve the existing gap in unstructured document parsing.

- The work reveals many valuable insights. For example, existing parsing techniques can fail in many irregular edge cases while smaller retrieval models could perform reasonably well in certain RAG applications.

- This work also studies the effectiveness of long-context LLMs compared to typical RAGs, which matches the current interest of the community.

**Additional Feedback:**

As written in the "Opportunities For Improvement" part.

**Clarity:**

In line 104 (Label collection), the author doesn’t clear state how they drive their dataset from the existing dataset. Which modification is performed to drive the each set from the existing dataset, e.g., from FINQA to FinHybrid?

**Documentation:**

Yes.

**Ethics:**

No.

**Limitations:**

No. It would be great that the author could add an explicit section to discuss the limitation.

**Opportunities For Improvement:**

For some datasets, e.g., PaperTab, the author doesn’t put effort to produce well parsed tables or text when it doesn’t exist in the original document. It would be great to put effort here and get well parsed content, which is essential for evaluating the parsing techniques.

**Relation To Prior Work:**

Yes.

**Summary And Contributions:**

This work proposed a benchmark for unstructured document analysis, collecting 3k real-world documents and 30k question-answer Paris. Unlike previous works that always assume clean or segmented inputs, this work considers end-to-end document analysis, from parsing unstructured document to answer generation. It also draws interesting insights by evaluating various parsing techniques and RAG models on the proposed benchmark.

---

> ### Author Rebuttal · Authors · 2024-08-16
>
> Thank you for your supportive and insightful comments.
>
> **Produce well-parsed content on PaperTab.**
>
> Acknowledging your valuable suggestion, we have additionally annotated 100 PaperTab tables into well-parsed Markdown texts, and we plan to provide more well-parsed content in the future. We assessed downstream tabular Q&A performance using these samples from PaperTab, with results shown in the following table.
>
> |            | Well Parsed | GPT-4-Omni | Raw Text |     CV   | CV + LLM |
> | ---------- | ----------- | ---------- | -------- | ---- | -------- |
> | GPT-4      | 42.8        | 44.3       | 42.4     | 38.6 | 40.7     |
> | Llama-3-8B | 35.8        | 37.7       | 36.5     | 34.6 | 32.1     |
>
> Echoing findings in Section 4.1 and Table 5, GPT-4-Omni and raw-text parsing show promising results, potentially surpassing well-parsed tables. This might be due to the fact that GPT-4-Omni and raw-text retain more structural cues such as indents and spacing, which might underscore key table elements for downstream LLM to understand.
>
> **Directly compare the parsed content.**
>
> We agree with the importance of direct comparisons of parsed content. We would like to clarify that our work focuses on LLM-based document understanding and the input parsed content is unstructured text. In this scenario, the quality of the content and the precise criteria for direct assessment are not well-defined; even manually annotated content may not significantly help LLM generation in some cases (as shown in the aforementioned table). We intend to explore and delve into the direct evaluation of document table parsing in future studies.
>
> **Add an explicit limitation section.**
>
> Thanks for the advice. We will add an explicit and detailed limitation section in the next version of our paper. The major limitation of our work can be summarized as: due to the evaluation and annotations that may be non-standard or non-scalable, some intricate analyses remain in future work, including direct comparison of parsed content, and in-depth analysis of noise sensitivity and hallucination.
>
> **"How they drive their dataset from the existing dataset."**
>
> Thanks for your valuable feedback. We mentioned the details of dataset construction in the supplementary material, and we will clarify it in the next version of our paper as follows :
>
> To provide the original documents for the collected Q&A labels, we first perform source-document identification, including retrieving, verifying, and collecting the complete document files. Then we enhance the data quality through the following transformations: (1) data cleaning by removing the Q&A pairs or documents lacking essential answers; (2) standardizing diverse data formats into consistently structured tables and JSON files, addressing the heterogeneity of presentation across different sub-datasets; (3) categorizing the queries in Qasper into table-centric and text-centric, thus forming the PaperTab and PaperText subsets for different application patterns; (4) converting html-token-based data type from NaturalQuestions into natural language, forming our user-friendly NqText dataset.

---

### Author Rebuttal · Authors · 2024-08-16

We thank all the reviewers for their time and valuable comments. We are encouraged by the reviewers' supportive feedback, recognizing that our work is valuable (snvC, FaEw, ubM5), comprehensive and thorough (FaEw, zcFM, ubM5), novel and beyond previous work (FaEw, snvC, zcFM) and well-written (snvC, FaEw, zcFM).

We attempt to address the suggestions and concerns of the reviewers with the following main updates:

* Produce additional well-parsed content in the PaperTab dataset, and conduct supplementary experiments, confirming our insights in parsing strategy.  [Reviewer snvC]
* Incorporate the LLM-based evaluator for a more comprehensive evaluation, further validating our pivotal findings. [Reviewer FaEw]
* Conduct experiments on LLM Q&A tasks without RAG, demonstrating the effectiveness of our datasets. [Reviewer zcFM]
* Add an explicit limitation section. [Reviewer snvC, FaEw]
* Clarify the utility of our dataset, and refine the explanation on some experiments. [Reviewer FaEw, zcFM, ubM5]
* Clarify more details of the dataset construction. [Reviewer snvC, zcFM]

---

### Decision · Program_Chairs · 2024-09-26

**Decision:**

Accept (Poster)

**Comment:**

The paper constructs Unstructured Document Analysis (UDA) benchmark to evaluate the performance of RAG. All reviewers agree that the benchmark is comprehensive and the evaluation is thorough, covering different RAG components. Please check the review comments for the details. The authors provides details responses to answer the review questions during the rebuttal phase, which should be reflected in the final version.